# Analysing State-Backed Propaganda Websites:
# a New Dataset and Linguistic Study

**Freddy Heppell, Kalina Bontcheva** and **Carolina Scarton**
Department of Computer Science, University of Sheffield, Sheffield, UK
{frheppell1, k.bontcheva, c.scarton}@sheffield.ac.uk

## Abstract

This paper analyses two hitherto unstudied sites sharing state-backed disinformation, Reliable Recent News (`rrn.world`) and WarOnFakes (`waronfakes.com`), which publish content in Arabic, Chinese, English, French, German, and Spanish. We describe our content acquisition methodology and perform cross-site unsupervised topic clustering on the resulting multilingual dataset. We also perform linguistic and temporal analysis of the web page translations and topics over time, and investigate articles with false publication dates. We make publicly available this new dataset of 14,053 articles, annotated with each language version, and additional metadata such as links and images. The main contribution of this paper for the NLP community is in the novel dataset which enables studies of disinformation networks, and the training of NLP tools for disinformation detection.

## 1 Introduction

Coordinated, state-backed disinformation operations have become an increasing problem in recent years, particularly surrounding the war in Ukraine (Morkūnas, 2022). In September 2022, a sophisticated network of *doppelganger* websites (impersonating genuine news sites from across Europe) was discovered by EUDisinfoLab (Alaphilippe et al., 2022) and later expanded on in a report from Meta (Nimmo and Torrey, 2022). Among these was also a small number of conventional false news sites.

The focus of this study is on two related disinformation sites in particular: Reliable Recent News[1] (RRN) and War On Fakes[2] (WoF). Both sites are multilingual, publishing in Arabic, Chinese, English, French, German, and Spanish, and RRN additionally in Italian[3]. They have been promoted by Russian government sources, including being shared by Russian embassies (Maitland,

2022; Roache, 2022), and publicised by the Ministry of Foreign Affairs of Russia's official Twitter account[4]. We focus on these two "news" sources due to their links to the Doppelganger network, their potential to deceive unsuspecting citizens (compared to better known propaganda sources such as Russia Today), and their prior exposure as disinformation spreaders (see Appendix A).

Backovic and Walter (2023) investigated the ownership of WarOnFakes, and stated it was operated by Russian journalist Timofey Vasiliev, a known affiliate of Russian propaganda groups, due to the presence of his name, email and phone number on the website. However, they do not state precisely how they found this information, and do not attempt to establish a link between Vasiliev and RRN or the Doppelganger operation.

Hanley et al. (2022) included selected articles from WarOnFakes and nine other disinformation websites in an analysis of narratives spread on Reddit. In contrast, our dataset includes all WarOnFakes posts and extracts the full article content.

Propaganda is defined as content that intentionally influences opinion to advance its creators' goals (Bolsover and Howard, 2017). Numerous propaganda datasets have previously been created, with both document-level (Rashkin et al., 2017; Barrón-Cedeño et al., 2019) and span-level (Da San Martino et al., 2019b) technique annotations, using articles collected from multiple disinformation sites. At article-level, classifiers using combinations of multiple linguistic representations based on style and readability outperform content representation (Barrón-Cedeño et al., 2019), whereas content-based transformer models such as BERT have seen use at span-level (Da San Martino et al., 2019a). Detectors are often evaluated on single datasets, prompting concerns on generalisation (Martino et al., 2020).

We are not aware of any prior work including RRN, nor of any work which has released a com-

---

[1] `https://rrn.world`, formerly called *Reliable Russia News* using rrussianews.com.

[2] `https://waronfakes.com`

[3] WoF also has a separate Russian-language site

[4] `https://twitter.com/mfa_russia/status/150022 3302941487107`

plete dataset of a disinformation operation, including a detailed linguistic analysis.

Thus the contributions of this paper are: i) a new publicly available[5] dataset of content from two state-backed disinformation websites; ii) a linguistic, topic, and temporal analysis of their articles; and iii) our open-source toolkit for processing site data and extraction of translations[6].

## 2 Methodology

### 2.1 Data Collection

In March 2023 we used the WordPress REST API[7] to obtain all posts from WoF and RRN. Each post was parsed to extract its text, removing non-article content (such as figure captions). The webpage of each post was then analysed to extract the different translations from the language picker. Our extraction tool supports the specific markup used by these two sites, but can be easily extended to support others. An example of an extracted article is shown in Appendix B.

Publication and modification times, which are provided in GMT by the API, were also converted to Moscow local time for analysis, since it is believed that at least one of the sites is based in Russia (Backovic and Walter, 2023).

### 2.2 Topic Analysis

The articles were clustered using BERTopic (Grootendorst, 2022). We assume that whilst each article may discuss many topics, each sentence of an article is likely to discuss a single topic. Articles were split using spaCy's dependency-parse-based sentenceizer, and sentences with less than 5 tokens were removed. The remaining sentences were embedded with Sentence Transformers MP-NET[8] (Reimers and Gurevych, 2019; Song et al., 2020). The dimensionality of each embedding was reduced using UMAP (McInnes et al., 2020) from 768 to 5, whilst keeping the structure of the higher-dimensional space. This is necessary to avoid the 'curse of dimensionality'[9].

The 5d embeddings were clustered with HDB-SCAN (Campello et al., 2013), which notably al-

lows for embeddings to not be included in a cluster, preventing overly broad clusters by forcing nearby but unrelated sentences in. It is expected that this produces a large number of outliers, since it is natural that many of the sentences in the articles will have meanings unrelated to any other. A minimum cluster size of 25 is set to prevent too many small clusters from being generated.

Keyword representations are generated by creating a bag-of-words vector of the unigrams and bigrams of each topic (excluding English stopwords) which is L1-normalized to account for cluster size. An adapted class-based TF-IDF is used to calculate the most significant words in each cluster. This representation is then fine-tuned by selecting keywords with a high Maximal Marginal Relevance, in order to maximise their diversity. The `diversity` parameter was set to 0.5. The top 3 most significant keywords are used to name the cluster.

Each article is then labelled with the unique set of clusters assigned to its sentences.

### 2.3 Article Backdating

In WordPress, article publication dates can be set to any given date, however this does not affect the auto-incrementing IDs which are generated in the order of article creation. Thus backdated articles can be detected based on their IDs being higher than that of their following articles, when they are ordered by supposed publication date.

### 2.4 n-gram Frequency

Frequent 2-4-grams were extracted using NLTK, after tokenisation, lowercasing, and stopword and punctuation removal. N-gram frequency was calculated monthly, and the most frequent 10 n-grams per month were selected, excluding the phrase "armed forces"[10], and n-grams which are part of another, longer n-gram of equal frequency (e.g. removing "ukrainian armed" in favour of "ukrainian armed forces"). We include ties for 10th place.

## 3 Analysis

### 3.1 Dataset Size and Coverage

Our dataset contains 14,053 translations of 3,447 articles posted between 4 Mar 2022 and 6 Mar 2023. Table 1 shows the number of articles per

---

[5]https://zenodo.org/records/10007383
[6]https://github.com/GateNLP/wordpress-site-extractor
[7]https://developer.wordpress.org/rest-api/
[8]https://huggingface.co/sentence-transformers/all-mpnet-base-v2 @ bd44305

[9]Clustering is difficult in higher dimensional spaces as distance is less meaningful (Aggarwal et al., 2001)

[10]This term is highly frequent, but is ambiguous as includes both Russian and Ukrainian armed forces, which appear as separate highly frequent trigrams.

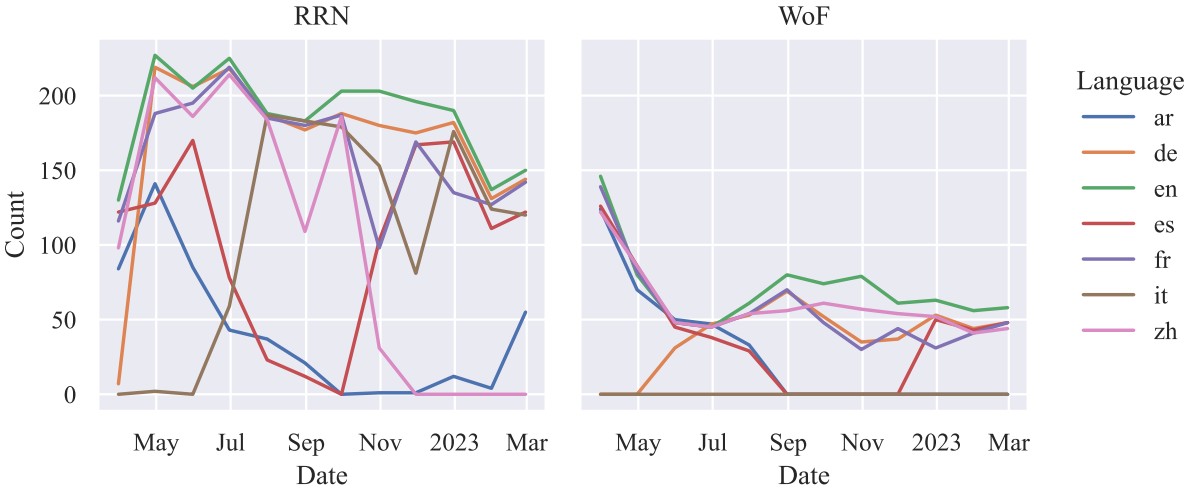

Figure 1: Monthly post counts across both sites, by reported article date. Partial data for March 2023 excluded.

| Language | Article Count | | | Mean #/article | |
| --- | --- | --- | --- | --- | --- |
| | RRN | WoF | Total | Toks. | Sents. |
| Arabic (ar) | 509 | 324 | 833 | 201.89 | 10.44 |
| German (de) | 2032 | 473 | 2505 | 339.90 | 15.91 |
| English (en) | 2265 | 864 | 3129 | 341.31 | 15.84 |
| Spanish (es) | 1229 | 468 | 1697 | 345.22 | 14.99 |
| French (fr) | 1968 | 683 | 2651 | 386.91 | 15.42 |
| Italian (it) | 1288 | - | 1288 | 429.89 | 19.03 |
| Chinese (zh) | 1220 | 730 | 1950 | 261.47 | 13.41 |
| All | 10511 | 3542 | 14053 | 338.89 | 15.37 |

Table 1: Number of articles per site, per language

site and language, and mean token and sentence counts[11].

## 3.2 Article Frequency

Figure 1 shows the proportion of each language over time for each site. The first WoF article is published on the 4th March 2022, and the first RRN article on the 11th. WarOnFakes has an unusual pattern of publication in its first few days, publishing sixty articles on the first day, and an average of 34 articles/day over the first 7 days, whereas RRN published only 7 articles on day one and an average of 21 articles/day over the first week.

Generally, posts are published on weekdays, with only 9.5% of posts having publication dates and 7.0% having modification dates on a Saturday or Sunday. The week beginning 2nd January 2023, much of which is public holidays in Russia[12], has

---

[11]Calculated using the spaCy tokeniser and rule-based sentenciser, with \n added to delimiters. Chinese segmented with PKUSEG web model (Luo et al., 2022). Arabic support is limited.

[12]https://www.cbr.ru/eng/other/holidays/

the lowest activity in the sites' history, with only 60 articles published on RRN and 19 on WoF. For comparison, the mean in other weeks is 200 (RRN) and 66 (WoF).

25 identical articles were published on both sites predominantly in March 2022, and in all but one case they were a WoF-style debunk. They were not published simultaneously on the two sites, nor is it consistent which site published first.

## 3.3 Language Coverage

Only a small minority of posts ($\sim$9.1%) are not available in English, and the majority of these do not have any translations at all, suggesting they are likely 'orphaned' translations. The mean number of available languages for a post is $4.1 \pm 1.5$ (1 std)

All site-language pairs continued to be published until the end of the collection period, except Arabic and Spanish on WoF and Chinese on RRN, which stopped in July and October 2022 respectively. Spanish posts resumed in December 2022.

## 3.4 Topics

Amongst the 45,991 English sentences in the English articles, 24,800 were considered outliers and 21,191 were assigned one of 144 topics. These topics ranged from broad, recurrent themes (e.g. #0, the donation of arms and aid to Ukraine) to more specific, time-limited ones (e.g. #139, the burning of the Quran by far-right activist Rasmus Paludan).

The mean number of topics assigned per article is $4.33 \pm 2.66$ (1 std). In the first week of the war in Ukraine (beginning 28th Feb 2022), the vast majority of articles are categorised as #2 (*russian*

*military, ukranian telegram, telegram channels, according [to] ukranian*). These articles are all from WarOnFakes (since RRN did not start publishing until the following week) and are claiming that various evidence from the war in Ukraine is fake.

Of the 144 topics we identified, 126 were assigned to articles from both RRN and WoF, and only 18 were assigned to posts from just one of the two sites. This demonstrates the significant topical overlap between the sites. Further details and figures are provided in Appendix C.

## 3.5 LIWC Analysis

We use LIWC2015 (Pennebaker et al., 2015) to compare the linguistic properties of English RRN and WoF posts against the metrics for genuine New York Times (NYT) articles provided by Pennebaker et al. (see Table 2 and Appendix C.1).

Emotional tone, which is on a scale of 0-100 (negative to positive), shows that RRN and WoF are written more negatively than real news, with WoF being even more negative than RRN. This is confirmed by the values for *Affective Processes*, which show that both sites use more emotion-laden words than NYT. The sub-metrics show this is skewed towards *negativity*, particularly *anger* (where both sites have over double the proportion of anger-indicating words than NYT).

All three sources focus most commonly on the present (e.g. words like "today", "is", "now"), however RRN and WoF do so at a higher rate than the NYT. RRN and WoF also use more future focus terms (e.g. "may", "will", "soon") compared to the NYT , and past focus terms (e.g. "ago", "did", "talked") less frequently. This suggests that the content of RRN and WoF comments is more speculative as compared to reputable journalism and is more focused on covering current events than past ones.

Table 3 shows the top 5 LIWC categories with the strongest correlation for each of the two sites. The strong correlation of colons and interrogatives for WoF is unsurprising, given its repeated use of the phrase "**What**'s really going on**:**". RRN's correlation with conjunctions suggests it tends to use more complex sentences. The remaining attributes are below the 0.3 threshold of strong correlation. However RRN is weakly correlated to personal pronouns which is due to its tendency to cover individual politicians (see Table 6 in Appendix C), while WoF is weakly correlated to impersonal pro-

| Metric | RRN | WoF | NYT |
|---|---|---|---|
| Tone | 27.71 ↓ | 15.06 ↓ | 43.61 |
| Affective Processes | 4.67 ↑ | 3.97 ↑ | 3.82 |
| Positive Emotion | 2.12 ↓ | 1.23 ↓ | 2.32 |
| Negative Emotion | 2.49 ↑ | 2.72 ↑ | 1.45 |
| Anger | 1.01 ↑ | 0.98 ↑ | 0.47 |
| Past Focus | 3.67 ↓ | 3.77 ↓ | 4.09 |
| Present Focus | 6.42 ↑ | 6.40 ↑ | 5.14 |
| Future Focus | 1.12 ↑ | 1.00 ↑ | 0.8 |

Table 2: Comparison of selected LIWC2015 attributes, compared to the New York Times

| RRN | | WoF | |
|---|---|---|---|
| Metric | $r$ | Metric | $r$ |
| Conjunctions | **0.311** | Colons | **0.487** |
| Pos. Emotions | **0.310** | Interrogatives | **0.373** |
| Pers. Pronouns | 0.277 | Impers. Pronouns | 0.293 |
| Discrepancies | 0.271 | See | 0.263 |
| Time | 0.270 | Leisure | 0.189 |

Table 3: Top 5 correlated LIWC values. **Bold** values above strength threshold.

nouns (i.e. one, you, they) as it tends to discuss groups, such as the Russian and Ukrainian armed forces (see Table 7 in Appendix C).

## 3.6 Article Backdating

Both sites tend to backdate non-English posts (by as much as 136 days in two cases, see Appendix C, Table 5), in order to make translations appear published at a similar time. The two most backdated articles are Spanish and Chinese translations of an English article, which were actually published 136 days later.

Our hypothesis for the backdating is due to limited resources articles were only translated into a given language when that became necessary for a particular disinformation campaign. In order to convey timeliness, the translations were then backdated to the date of the original.

## 3.7 n-gram Analysis

Tables 6 and 7 in Appendix C show the top occuring n-grams per month for the respective websites. The most frequent "really going" n-gram on WoF is part of the phrase "What's really going on", which appears in all of its fact-check-style articles. The n-gram also appears frequently in the first month of RRN data, due to the articles copied from WoF.

| Category | RRN | WoF |
|---|---|---|
| Accidental Cyrillic | 58 | 34 |
| Forgotten Cyrillic | 15 | 25 |
| Intentional | 36 | 10 |
| Unclear | 0 | 2 |

Table 4: Frequency of Cyrillic usage reasons

On WoF, the most frequent n-grams typically relate directly to the war in Ukraine itself ("russian troops", "ukranian armed forces"), whereas on RRN they relate to the consequences of the conflict for the rest of the world ("united states", "russian gas"). Consequently, the most frequent n-grams on WoF are relatively constant across the different months, whereas RRN's n-grams change from one month to the next as they tend to be connected to current affairs. For example, the bigram "anti-russian sanctions" enters the top 10 in June 2022, and remains the second most used bigram from July to September, and refers to the damage allegedly caused to Western economies. Other terms demonstrate that RRN also covers some genuine news, e.g. "elizabeth ii" in September 2022 and "world cup" in November and December 2022.

Even though to a much lesser degree, WoF still responds to specific highly controversial events from the conflict. For example in August 2022, in response to Ukraine and Russia blaming each other for the shelling of the Zaporizhzhia nuclear power station[13], the n-grams "nuclear power" and "nuclear power plant" both appear with high frequency in WoF articles that promote the Russian perspective on these events.

### 3.8 Presence of Cyrillic Characters

178 of the articles were found to contain characters in the Cyrillic codepoint range (Table 4), which were manually examined to determine the reason.
**Accidental Cyrillic**: Incorrect usage of Cyrillic characters instead of the intended character in the Latin alphabet. For example, 11 times the "c" in Robert Habeck, a German politician, is actually the identical-looking lowercase Cyrillic Es[14].
**Forgotten Cyrillic**: Issues with translation where a Russian sentence was left in the article, with or without the target language translation.
**Intentional**: Expected usage of Cyrillic characters

e.g. the name of a Russian organisation.
**Unclear**: We were unable to determine why the characters were used.

Given that both RRN and WoF had forgotten Russian text in all languages, we hypothesise that all articles were originally written in Russian. Two Arabic articles on RRN contain the phrases *"the translation is too long"* and *"save translation"* in Russian, likely copied from a machine translation tool's UI, although we were not able to determine the specific tool used. Although this was only found in one language on one of the sites, it suggests it is more likely the articles are machine than human translated.

## 4 Future Work

There is much additional work which could be performed on this dataset. Although we identify the subject of articles via topic clustering and n-grams, we do not attempt to identify stance towards it. More complex topic analysis, such as identifying commonly co-occuring topics, would also be possible. Given the mixture of true and false posts on the sites, this dataset may be a useful resource for automated fact-checking, although this would require human annotation and ground-truth may be difficult to establish in the complex information environment of the war in Ukraine.

## 5 Conclusion

This paper presented an analysis of the Russian disinformation sites Recent Reliable News and WarOnFakes, including an analysis of the articles' topics, publication times, and linguistic properties. We show that the sites cover a diverse range of topics, and that their linguistic properties differ from those of reputable media. We analysed the presence of Cyrillic characters due to site operator errors, and their practice of backdating articles, showing that a significant proportion of translations are falsely dated. This new multilingual dataset will facilitate further research in disinformation analysis and promote repeatability.

### Limitations

Although our work provides a complete collection of WoF and RRN, since these two websites seem to be highly related, it is unsurprising that they tend to publish similar types of content. Therefore this dataset cannot be considered fully representative

---

[13]https://reut.rs/46KWvTS
[14]https://en.wikipedia.org/wiki/Es_(Cyrillic)

of all kinds of Russian disinformation. Nevertheless, it is complementary to overtly Russian state media, such as Sputnik and Russia Today. Unfortunately, due to the ban on accessing their content from the EU, we could not supplement the dataset from those sources or compare against them.

Our topic analysis model has not been formally validated, for example by comparing topics to those assigned by human or expert annotators. Some small scale manual validation was performed in order to find good hyperparameters, however this consisted of inspecting a small random sample of some of the categories. A particular area warranting validation in future work is examining the texts not assigned categories. These are only a very small number, however as we aggregate sentence classifications at article level, which means that an article can be assigned the correct topics even if some of its sentences may not be.

In our LIWC analysis, we compare to the New York Times data provided by Pennebaker et al. (2015). Although this is the closest source out of the provided LIWC baselines, the New York Times represents a more formal style of journalism than many online media. In future work we plan to compare these two disinformation sites against official state-affiliated news sources such as Russia Today.

Finally, we did not analyse the separate Russian-language edition of WarOnFakes. As it is a separate site in Russian only, there is no reliable way to connect its articles to their similar English-language versions (if such are published). Analysing the Russian WoF website is planned for future work, as it requires adaptation of the analysis to be bilingual, which is out of scope for this paper.

## Ethics

The data collection was carried out in accordance with our institutional ethics policy.

Collection was via the Wordpress API, followed by automated processing and a limited amount of manual analysis by the authors. No external volunteers or crowd-workers were recruited. Due to the disinformation nature of these two websites, the data may contain content which is disturbing or distressing. Therefore we limited the possibility of harm during analysis by: i) minimising the number of individual articles studied by the authors as much as possible; ii) where necessary, viewing only the text of articles, to avoid the possibility of viewing distressing media; iii) ensuring familiarity with supporting resources for researchers working with potentially disturbing content.

As the websites in question are not legitimate news websites, they do not have a terms of use to allow or prohibit the acquisition of their content. We consider the collection and distribution of their articles in the public interest, due to the prominence of their disinformation and the harm that results from it. It is not feasible to contact them to obtain permission, as they have previously been unresponsive to enquiries [15]. The dataset does not include images, as in many cases they appear to have been taken from stock agencies. This is a commonly used tactic by disinformation websites.

We have checked that the dataset does not contain personally identifiable information in the user data files, as all users have either generic (e.g. "Admin") or random (e.g. "UiXnZyvH") names. No user comments were available to collect.

It is possible that the process of creating a disinformation dataset increases the spread and prominence of the disinformation. We would argue that is not the case with this dataset as we: i) are only focusing on content from disinformation websites, the low credibility of which has already been widely publicised (see Appendix A); iii) are not increasing the longevity of disinformation narratives by preserving them after they have being taken down, since the two independent websites that are publishing them are still publicly accessible via all common search engines.

Some articles make reference to individuals, albeit only public figures to our knowledge, and many contain narratives which are hateful towards individuals and groups. We encourage researchers who use this dataset to do so responsibly, and in particular to avoid highlighting specific individuals and to ensure that the disinformation narratives are presented alongside authoritative evidence of their untrue nature. We would like to specifically discourage the use of this dataset for training generative models that are capable of creating new disinformation. The dataset is released under a license which prohibits commercial activity.

## Acknowledgments

This work is partially supported by the UK's innovation agency (InnovateUK) grant number

---

[15]NewsGuard attempted to contact them as part of their review process (Maitland, 2022; Roache, 2022)

10039039 (approved under the Horizon Europe Programme as VIGILANT, EU grant agreement number 101073921).[16] Freddy Heppell is supported by a University of Sheffield Faculty of Engineering PGR Prize Scholarship.

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

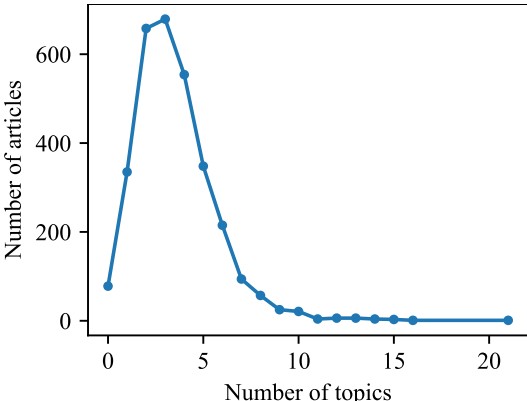

Figure 2: Number of unique topics assigned per article

| Language | Backdated | Mean | Max |
|----------|-----------|------|------|
| ar | 39.4% | -8d | -34d |
| es | 11.5% | -9d | -136d |
| de | 11.1% | -7d | -104d |
| fr | 10.2% | -9d | -109d |
| zh | 8.4% | -15d | -136d |
| it | 6.9% | -4d | -26d |
| en | 0.2% | -1d | -2d |

Table 5: Backdating per language for both sites.

## A    Evidence of Disinformation

For WarOnFakes, there is a substantial number of articles and fact-checks establishing it as a disinformation source. PolitiFact undertook a review of over 380 of their fact-checks and found a significant number of falsehoods[17]. In an article by AFP via France24, Roman Osadchuk, from the Atlantic Council's Digital Forensic Research Lab (DFRLab), is quoted as saying "Since Russia's invasion, the 'War On Fakes' initiative has become a powerhouse of spreading false debunks" and "It is an effective tool of state propaganda and disinformation" [18]. The Institute of Network Cultures describes it as "Kremlin-Sponsored Particpatory Propaganda"[19], and highlights connections between the Russian state and the website, including promotion from organisations under the Russian Ministry of Foreign Affairs, and on the Russian Ministry of

Defence's Telegram channel. BBC Monitoring, the specialist media source analysis division of BBC News, states "Some of its fact-checks are genuine but most content is Russian talking points on the invasion which do not stand up to scrutiny"[20].

The site has also been covered by EUvsDisinfo[21], DFRLab[22], the European Digital Media Observatory[23], and Media Bias/Fact Check[24].

RRN has received comparatively less attention from fact checkers, however was described as disinformation by NewsGuard (Maitland, 2022), which additionally claims that they reuse content from WarOnFakes, and EU Disinfo Lab have noted a connection in the hosting infrastructure of the two sites (Alaphilippe et al., 2022). It is therefore probable that the apparent state-backing of WarOnFakes also applies to RRN.

## B    Data Example

Figure 3 shows an example of an article published on WoF[25] in English, French, Spanish, Chinese and Arabic. Full texts are omitted for languages other than English. This story was judged to be fake by fact-checkers[26]. Usage of guillemets (« ») as quote marks is reproduced as returned by the WordPress API, but this appears to be normalised when the page is rendered.

## C    Detailed Dataset Statistics

Figure 4 shows a weekly chart of the 10 most common topics on the site. In general, there is no clear variation between these topics, with the exception of the initial popularity of the topic #2 due to the majority of posts that week being from WarOnFakes. The significant dip in January 2023 is due to the Russian public holidays discussed in section 3.2.

Figure 2 shows the distribution of sentence-level topic counts aggregated for each article. 78 posts were not assigned any topic, the majority of articles

---

[17]https://www.politifact.com/article/2022/aug/08/how-war-fakes-uses-fact-checking-spread-pro-russia/

[18]https://www.france24.com/en/live-news/20230216-fake-fact-checks-seek-to-obscure-russian-role-in-war

[19]https://networkcultures.org/tactical-media-room/2022/07/22/weaponized-osint-the-new-kremlin-sponsored-participatory-propaganda/

[20]https://monitoring.bbc.co.uk/product/c203aqg1
[21]https://euvsdisinfo.eu/riding-the-bomb/
[22]https://medium.com/dfrlab/russian-telegram-channel-embraces-fact-checking-tropes-to-spread-disinformation-c6a54393c635
[23]https://edmo.eu/2022/03/17/russian-propaganda-disguising-as-fact-checking-a-statement-from-the-edmo-taskforce/
[24]https://mediabiasfactcheck.com/war-on-fakes-bias/
[25]https://waronfakes.com/civil/fake-russian-aviation-struck-a-maternity-hospital-with-mothers-and-children/
[26]https://reut.rs/3tEvFfk

have 1-3 topics, however in the extreme some have as many as 21 - this is an article from WarOnFakes *"What happened in Bucha? A full analysis of the Ukrainian provocation"* , a long article supposedly explaining the truth about many elements of the Bucha massacre.

Table 5 shows the proportion of backdated articles per language, and the mean and maximum backdating period for each. For English, a small number of posts are backdated after a short period of time. It is likely this is caused by posts that have been forward-dated (i.e. set to be published in the future) by one or two days, resulting in subsequent posts appearing to be backdated until the publication date catches up. However, for other languages, backdates are for a much longer period.

## C.1 Complete LIWC2015 Data

The complete listing of LIWC2015 is included in Table 8, in the hope it can be used for comparison in future work.

ENGLISH

**Fake: Russian aircraft attacked a maternity hospital with mothers and children inside**

What is fake about:

Information that Russia launched an airstrike on a maternity hospital in Mariupol is being spread online. Ukrainian President Volodymyr Zelensky called it «an atrocity» and said that women and children remained under the rubble.

The fact

Despite the fact that information about the strike appeared in the middle of the day of March 8, no single patient was visible on numerous videos and photos. The footage of pregnant women appeared on the Internet much later – in the evening of March 9. However, it immediately was circulated by all news agencies, social media, popular communities and bloggers, which may be the result of a preplanned campaign. Moreover, it was happening despite the fact that the locals themselves claimed that there were no patients or members of the staff in the maternity hospital.

This story is rather dubious. It is logical to assume that if there really had been patients then the rescue service officers and eyewitnesses who arrived at the scene would immediately have taken photos of the accident scene with their phones, without waiting for a well-known photographer. However, it so happened that the well-known Ukrainian propaganda activist Evgeniy Maloletka was the first to prepare and publish the photographs.

Today we received indisputable confirmation that the «photos of pregnant women" were staged. The Ukrainians used a model called Marianna who comes from Mariupol for the most striking photos (there are three in total). It is notable that she played roles of two different pregnant women at the same time: she even had to change clothes and the color of her hair, which, however, is not surprising: in fact, Marianna is a well-known beauty blogger in the region. It's worth noting that the girl is indeed pregnant, but she just could not have been in the maternity hospital: the Azov militants had used the medical facility for several days as a fortified stronghold that does not function as a maternity hospital any longer. The main heroine of this hoax has already been caught in the spotlight. In the comment section of her Instagram account there are already more than 500 comments under her last post written by real users condemning the girl for participating in information manipulations.

FRENCH

**L'infox: les forces aériennes russes ont bombardé la maternité. Les femmes et les enfants ont été ciblés**

SPANISH

**Fake: La aviación rusa atacó al hospital materno-infantil con madres y bebés**

CHINESE

假新闻：俄罗斯空军袭击了产科医院

ARABIC

**خبر مزيف: الطيران الروسي هاجم مستشفى الولادة بها الأمهات والأطفال**

Figure 3: An example of an article from WoF. Article text is omitted for non-EN languages for space.

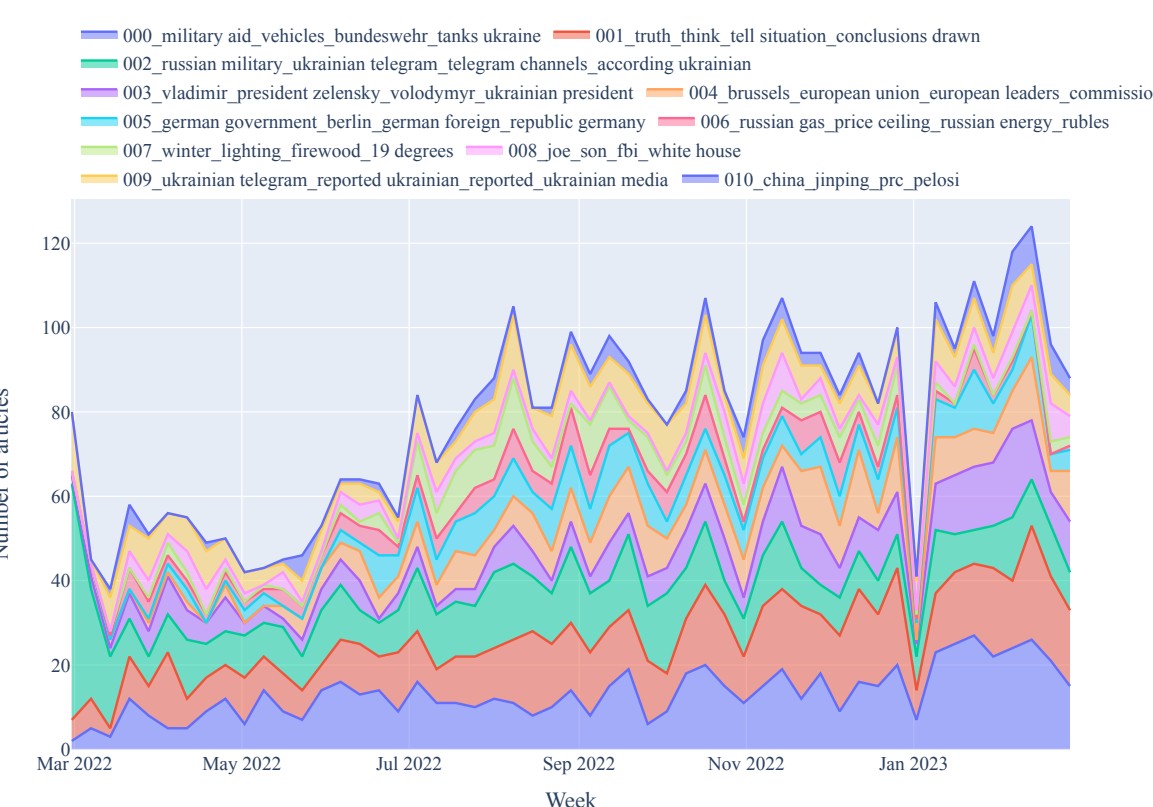

Figure 4: Weekly frequency of top 10 clusters. Partial data for March 2023 excluded.

| Mar 22 | Apr | May | Jun | Jul | Aug |
|---|---|---|---|---|---|
| united states | united states | united states | united states | prime minister | prime minister |
| russian gas | russian federation | special operation | european union | anti-russian sanctions | anti-russian sanctions |
| russian federation | russian military | prime minister | anti-russian sanctions | european union | nord stream |
| really going | special operation | russian military | prime minister | nord stream | energy crisis |
| russian military | ukrainian armed | ukrainian armed | sanctions russia | ukrainian refugees | russian gas |
| european countries | ukrainian armed forces | ukrainian military | white house | ukrainian army | united states |
| prime minister | le pen | western countries | joe biden | von der leyen | per cent |
| ministry defense | russian troops | ukrainian armed forces | ukrainian refugees | european commission | nuclear power |
| people republic | russian gas | ukrainian soldiers | european commission | german government | energy prices |
| ukrainian nationalists | | joe biden | olaf scholz | united states | nord stream 2 |
| | | russian oil | ukranian crisis | | |

| Sep | Oct | Nov | Dec | Jan 23 | Feb |
|---|---|---|---|---|---|
| prime minister | prime minister | united states | united states | united states | united states |
| anti-russian sanctions | united states | joe biden | world cup | prime minister | white house |
| nord stream | liz truss | prime minister | white house | ukrainian army | joe biden |
| energy crisis | nuclear power | white house | joe biden | ukrainian armed | prime minister |
| united states | vladimir putin | world cup | prime minister | ukrainian armed forces | vladimir zelensky |
| european commission | nord stream | foreign minister | ukrainian army | white house | military aid |
| vladimir putin | anti-russian sanctions | elon musk | vladimir putin | foreign minister | nord stream |
| elizabeth ii | crimea bridge | vladimir putin | emmanuel macron | joe biden | ukrainian armed forces |
| nuclear power | energy crisis | rishi sunak | price cap | foreign policy | last year |
| liz truss | white house | anti-russian sanctions | elon musk | olaf scholz | ukrainian army |
| russian gas | | | new year | world war | world war |
| | | | ordinary people | | |

Table 6: Top n-grams for RRN

| Mar 22 | Apr | May | Jun | Jul | Aug |
|---|---|---|---|---|---|
| really going | russian troops | really going | really going | really going | really going |
| fake message | telegram channels | telegram channels | telegram channels | telegram channels | ukrainian armed forces |
| telegram channels | forces ukraine | russian military | ukrainian telegram channels | ukrainian armed forces | telegram channels |
| russian military | fake news | ukrainian telegram | shopping center | saudi arabia | power plant |
| forces ukraine | armed forces ukraine | ukrainian telegram channels | ukrainian armed forces | fake russian | fake russian |
| armed forces ukraine | fake message | russian troops | fake russian | ukrainian telegram channels | ukrainian telegram |
| ministry defense | russian soldiers | special operation | russian military | russian armed forces | nuclear power |
| russian federation | really going | ukrainian armed forces | according ukrainian | russian military | ukrainian telegram channels |
| ukrainian telegram channels | russian military | fake ukrainian | fake according | western media | nuclear power plant |
| russian armed | ukrainian telegram channels | russian armed forces | fake ukrainian | ukrainian side | russian armed |
| russian armed forces | | ukrainian sources | russian troops | | russian armed forces |
| | | | ukrainian media | | ukrainian side |

| Sep | Oct | Nov | Dec | Jan 23 | Feb |
|---|---|---|---|---|---|
| really going | really going | really going | really going | really going | really going |
| telegram channels | ukrainian armed forces | telegram channels | ukrainian armed forces | united states | telegram channels |
| ukrainian armed forces | telegram channels | ukrainian army | ukrainian army | telegram channels | military operation |
| ukrainian telegram | russian armed forces | ukrainian telegram | russian armed forces | ukrainian telegram | special military operation |
| ukrainian telegram channels | russian federation | ukrainian telegram channels | telegram channels | ukrainian army | russian federation |
| fake russian | air defence | channels really going | vladimir zelensky | ukrainian telegram channels | ukrainian telegram channels |
| channels really going | ukrainian telegram channels | russian armed forces | russian federation | air defence | telegram channels really going |
| russian armed forces | vladimir putin | ukrainian media | telegram channel | russian armed forces | united states |
| telegram channels really | fake russian | air defence | president vladimir | ukrainian armed forces | nord stream |
| telegram channels really going | ukrainian army | telegram channels really going | war fakes | ukrainian propaganda | fake russian |
| | | | | | vladimir putin |

Table 7: Top n-grams for WoF

| Category | RRN | WoF | All |
|---|---|---|---|
| Word count (mean) | 313.80 | 249.24 | 295.74 |
| **Summary Variables** | | | |
| Analytic | 94.37 | 95.12 | 94.58 |
| Clout | 63.62 | 56.75 | 61.70 |
| Authentic | 23.13 | 22.95 | 23.08 |
| Emotional Tone | 27.71 | 15.06 | 24.17 |
| **Language Metrics** | | | |
| Words/sentence | 19.92 | 20.15 | 19.98 |
| Words > 6 letters | 27.97 | 29.12 | 28.30 |
| Dictionary words | 76.81 | 75.33 | 76.40 |
| **Function Words** | 45.94 | 46.86 | 46.20 |
| Total pronouns | 6.30 | 6.20 | 6.27 |
| Personal pronouns | 2.70 | 1.54 | 2.38 |
| 1st pers singular | 0.16 | 0.06 | 0.14 |
| 1st pers plural | 0.50 | 0.38 | 0.46 |
| 2nd person | 0.16 | 0.09 | 0.14 |
| 3rd pers singular | 0.95 | 0.41 | 0.80 |
| 3rd pers plural | 0.94 | 0.60 | 0.84 |
| Impersonal pronouns | 3.59 | 4.66 | 3.89 |
| Articles | 10.30 | 11.06 | 10.51 |
| Prepositions | 15.81 | 16.09 | 15.89 |
| Auxiliary verbs | 6.53 | 7.21 | 6.72 |
| Adverbs | 3.23 | 3.83 | 3.40 |
| Conjunctions | 4.44 | 3.37 | 4.14 |
| Negations | 1.17 | 1.16 | 1.17 |
| **Other Grammar** | | | |
| Common verbs | 11.01 | 11.03 | 11.02 |
| Common adiectives | 3.97 | 3.62 | 3.87 |
| Comparisons | 2.02 | 1.46 | 1.86 |
| Interrogatives | 0.92 | 1.66 | 1.13 |
| Number | 2.13 | 1.75 | 2.02 |
| Quantifiers | 1.62 | 1.24 | 1.51 |
| **Psychological Processes** | | | |
| Affective processes | 4.67 | 3.97 | 4.47 |
| Positive emotion | 2.12 | 1.23 | 1.87 |
| Negative emotion | 2.49 | 2.72 | 2.55 |
| Anxiety | 0.38 | 0.22 | 0.34 |
| Anger | 1.01 | 0.98 | 1.00 |
| Sadness | 0.37 | 0.21 | 0.33 |
| Social processes | 6.82 | 4.84 | 6.26 |
| Family | 0.15 | 0.09 | 0.13 |
| Friends | 0.16 | 0.09 | 0.14 |
| Female references | 0.34 | 0.19 | 0.30 |
| Male references | 0.85 | 0.41 | 0.72 |
| Cognitive processes | 8.26 | 7.69 | 8.10 |
| Insight | 1.52 | 1.36 | 1.48 |
| Causation | 1.72 | 1.91 | 1.77 |
| Discrepancy | 0.96 | 0.47 | 0.82 |
| Tentative | 1.36 | 1.13 | 1.30 |
| Certainty | 1.11 | 1.02 | 1.09 |
| Differentiation | 2.34 | 2.26 | 2.32 |
| Perceptual processes | 1.72 | 2.03 | 1.81 |
| See | 0.65 | 1.32 | 0.84 |
| Hear | 0.63 | 0.42 | 0.57 |
| Feel | 0.34 | 0.22 | 0.30 |
| Biological processes | 1.24 | 1.06 | 1.19 |
| Body | 0.40 | 0.36 | 0.39 |
| Health | 0.56 | 0.57 | 0.57 |
| Sexual | 0.04 | 0.03 | 0.04 |
| Ingestion | 0.27 | 0.14 | 0.24 |
| Drives | 8.41 | 6.95 | 8.00 |
| Affiliation | 1.57 | 1.25 | 1.48 |
| Achievement | 1.54 | 1.06 | 1.40 |
| Power | 4.60 | 4.05 | 4.44 |
| Reward | 0.77 | 0.51 | 0.70 |
| Risk | 0.97 | 0.64 | 0.88 |
| Time orientations | | | |
| Past focus | 3.67 | 3.77 | 3.70 |
| Present focus | 6.42 | 6.40 | 6.42 |
| Future focus | 1.12 | 1.00 | 1.09 |
| Relativity | 13.77 | 13.05 | 13.57 |
| Motion | 1.70 | 1.78 | 1.72 |
| Space | 7.73 | 8.02 | 7.81 |
| Time | 4.39 | 3.21 | 4.06 |
| Personal Cocnerns | | | |
| Work | 3.77 | 3.32 | 3.64 |
| Leisure | 0.76 | 1.29 | 0.91 |
| Home | 0.34 | 0.31 | 0.33 |
| Money | 1.48 | 0.67 | 1.25 |
| Religion | 0.38 | 0.35 | 0.37 |
| Death | 0.38 | 0.43 | 0.39 |
| Informal Language | 0.23 | 0.19 | 0.22 |
| Swear words | 0.01 | 0.01 | 0.01 |
| Netspeak | 0.06 | 0.09 | 0.07 |
| Assent | 0.04 | 0.04 | 0.04 |
| Nonfluencies | 0.08 | 0.06 | 0.07 |
| Fillers | 0.02 | 0.00 | 0.02 |
| **Punctuation** | | | |
| Total Punctuation | 15.09 | 13.87 | 14.75 |
| Periods | 5.26 | 5.31 | 5.28 |
| Commas | 4.91 | 4.14 | 4.70 |
| Colons | 0.36 | 1.08 | 0.56 |
| Semicolons | 0.05 | 0.03 | 0.04 |
| Question marks | 0.13 | 0.03 | 0.11 |
| Exclamation marks | 0.09 | 0.01 | 0.07 |
| Dashes | 0.69 | 0.70 | 0.69 |
| Quotation marks | 2.09 | 1.52 | 1.93 |
| Apostrophes | 0.97 | 0.57 | 0.86 |
| Parentheses | 0.25 | 0.37 | 0.28 |
| Other punctuation | 0.28 | 0.12 | 0.23 |

Table 8: Complete LIWC2015 listings