# OpenReview forum: "Analysing State-Backed Propaganda Websites: a New Dataset and Linguistic Study"
_EMNLP/2023/Conference — EMNLP 2023 Main_

### Official Review · Reviewer_q5Fz · 2023-07-26

**Soundness:** 3

**Excitement:**

4: Strong: This paper deepens the understanding of some phenomenon or lowers the barriers to an existing research direction.

**Missing References:**

There is no background analysis. There is much work done in propaganda detection, both in the publication of linguistic resources, such as datasets and in the development of models to detect it. In addition, propaganda has been studied for many years in other disciplines. All this background needs to be accounted for in a paper that presents a new resource for automatic propaganda detection.

**Paper Topic And Main Contributions:**

This paper treats an interesting topic and introduces a valuable, very timely resource, namely a publicly available dataset of state-backed propaganda, enriched with linguistic, topic and temporal analysis. They also share an open-source toolkit for processing information from the source site and translations, such as n-gram frequency, distribution of languages in which the articles are translated or articles shared by the two sites.

**Questions For The Authors:**

A) Something that remains unclear to me is how are the articles labelled with topic, as it is the sentences the ones which are clustered. The article mentions that “Each article is then labelled with the unique set of clusters assigned to its sentences.”, so I am assuming that each article has a different number of labels and that some articles might actually cover a high number of topics.

B) The dataset contains 3,447 articles, which, translated into different languages, make 14,053. Those original 3,447 articles, are they in the same language? I can’t find that figure in Table 1, so I understand they are originally written in different languages, is that right? Do these sites have an official or primary language and translations in others or articles can be written, originally, in different languages?

C) I feel the topic detection experiment design might have some issues, if more than half of the sentences of a dataset are considered outliers. Could you explain this situation?

**Reasons To Accept:**

This is an interesting paper. I think the dataset can be very valuable not only for the NLP community, but also for analysis from other disciplines, such as discourse analysis, sociology, politics or journalism.  I found the analysis interesting, although more from a sociological, cultural perspective than from an NLP point of view.

**Reasons To Reject:**

There is no background analysis. There is much work done in propaganda detection, both in the publication of linguistic resources, such as datasets, and in the development of models to detect it. In addition, propaganda has been studied for many years from other disciplines. All this background needs to be accounted for in a paper that presents a new resource for automatic propaganda detection.

Examples of the dataset would be very helpful in a paper that introduces a new dataset, as the reader learns about your work, but is not able to see it.

**Reproducibility:**

4: Could mostly reproduce the results, but there may be some variation because of sample variance or minor variations in their interpretation of the protocol or method.

**Reviewer Confidence:**

4: Quite sure. I tried to check the important points carefully. It's unlikely, though conceivable, that I missed something that should affect my ratings.

**Typos Grammar Style And Presentation Improvements:**

Some small issues I detected that you might want to consider:

- Line 98-99: This is necessary to avoid the ‘curse of dimensionality’. → Needs citation and explanation
- Table 1: Which are the languages? Not all abbreviations are easy to understand for me.
- 2.4. n-gram Frequency → I assume this is for each website, but it is not specified.
- Section 3.2. refers to Figure 2, which is not included in the paper, but in the annex. Same with Table 6.
- 2.3 and 3.6. are both “Article backdating”

---

> ### Author Rebuttal · Authors · 2023-08-28
>
> We acknowledge that the background information does not include analysis of the propaganda detection field. Given limited space, we decided to focus on analysing prior work on these specific sites and comparing our work to it. Although there are prior datasets for propaganda detection, we are not aware of any that are parallel multilingual, and thus limiting direct comparisons. We believe a survey of propaganda detection techniques would not be out of this paper's scope, as it does not itself propose a method. However, we agree that the paper would be improved with an overview of applicable problems, such as propaganda detection, which would include an explicit comparison to prior datasets. We will add this to the paper.
>
> We will add an example of an article to better illustrate the type of content.
>
> Thank you for your grammar corrections and suggestions for clarification, we will address these.
>
> ---
>
> **A**: This is correct, a given article’s labels (0 or more) are the combined set of its sentences labels (1 per sentence). Most articles had 1-3 topics assigned, but a small number were assigned many more  (see figure 1) -- typically very long articles that appear to have been comprised of previously published articles. We will clarify this in the paper.
>
> **B**: Around 60% of the 3,447 articles were first published in English, and around 30% were published in English as a subsequent language. Of the ~10% that were never published in English, they almost all have no translations set, some were manually checked and the translations do exist but were mistakenly not linked by the site operators.
>
> The fact that nearly all articles are available in English and that it is the default language on both sites, suggests that they intend English to be the primary language. However, given the presence of Cyrillic characters in some English articles that are most likely to be caused by mistakes when translating, it appears that at least some of the English content is originally written in Russian.
>
> Please note that, whilst reviewing statistics for this answer, we identified an error in the original version of the paper. On line 172 it states 2.4% of posts are not available in English. This figure was calculated incorrectly, 2.4% refers to the percentage of the 14,053 translations that do not have an English version. It should be the percentage of the 3,447 articles that do not have an English version, which is 9.1%. We will correct this in the paper, as well as add the above clarifications.
>
> **C**: Only very short sentences are removed before clustering so some of the sentences will be irrelevant or contain little useful information, and some sentences may be on specific topics that are too distinct to cross the 25 size threshold to form a cluster. Whilst it is possible to assign outliers to nearby clusters, or further tweak clustering parameters to minimise them, this may force outliers into irrelevant topics or cause the creation of too many extremely specific small topics and a reduction of broader, more useful topics.
>
> As we aggregate sentence-level topics at article level, the presence of outliers at sentence level is less significant. Intuitively, an article on a given topic will have many ‘chances’ to be assigned correctly, as it will have many sentences on that topic. It is better for a sentence to incorrectly be considered an outlier (because another sentence in the same article will likely be correctly classified) than to be assigned to the wrong cluster (because the whole article’s labels will include this incorrect cluster).
>
> When aggregated, only 78 articles were outliers (i.e. all sentences within the article were outliers). We manually reviewed these articles: many are extremely short texts where the article was primarily a video or image, require external knowledge to interpret, or are on specific one-off news stories.

---

### Official Review · Reviewer_Rv3e · 2023-08-03

**Soundness:** 3

**Excitement:**

4: Strong: This paper deepens the understanding of some phenomenon or lowers the barriers to an existing research direction.

**Missing References:**

It could be interesting to mention or use Scattertext (Kessler 2017) to compare the sources.

**Paper Topic And Main Contributions:**

The article introduces a corpus on state-backed disinformation, provides a quantitative overview of the resource, and evaluates different detection approaches.

**Questions For The Authors:**

Question A: Other than backdating, did you work with other metadata like author names, categories and tags, or titles?

Question B: You mention holidays in section 3.2, did you witness other overlapping holidays?

Question C: The article lacks a section on future work, yet other text analyses are possible, do you have experience with language models for example?

**Reasons To Accept:**

Scientifically sound article, relevant cues on disinformation, interesting discussion on the methods and their limitations, as well as on ethical issues.

Decisions made on the tools are clearly described and quite valuable.

**Reasons To Reject:**

Part of content analysis is specific to the websites of interest and cannot be generalized easily.

Beyond topic analysis which involves a great deal of fine-tuning and may not be easy to reproduce on other datasets, the article lacks more refined textual analysis, i.e. proper sentiment detection other than word lists (section 3.5) and better word grouping or filtering methods than n-gram analysis (section 3.7).

**Reproducibility:**

5: Could easily reproduce the results.

**Reviewer Confidence:**

3: Pretty sure, but there's a chance I missed something. Although I have a good feel for this area in general, I did not carefully check the paper's details, e.g., the math, experimental design, or novelty.

**Typos Grammar Style And Presentation Improvements:**

Line 099: you could explain the curse of dimensionality for the sake of clarity.

Line 217: sub metrics → sub-metrics?

The sections on backdating (2.3 and 3.6) could be merged into a single one. Information in Table 5 on article backdating could be added to the article if there is enough room.

Generally speaking, sections on article text and article metadata could come in groups to make the results easier to follow.

---

> ### Author Rebuttal · Authors · 2023-08-28
>
> It’s unclear which part of the content analysis the reviewer is referring to, but we believe that our analysis techniques are generalisable to other sites (e.g. the n-grams and topics). The only part that may be difficult to generalise is the article backdating detection, which requires the site to use a platform that in some way ‘leaks’ the creation order of articles. However, this will work on any site that is powered by the WordPress content management system, which is widely used in general, and appears popular for disinformation sites due to its ease of setup. Our tool to extract translations currently supports the internationalisation plugin used by the sites, but is designed to be easily extended to others. The Cyrillic detection is the part that is most specific to these sites due to their place of origin, but similar ‘unicode traps’ exist for other languages.
>
> In our experience, the topic modelling process was in fact relatively straightforward. The library is well documented and in our experience did not require significant tuning of hyperparameters, particularly as the HDBSCAN algorithm automatically determines the number of clusters to make. It would be, however, difficult to compare topics across datasets, as the topics that form are dependent on a vast number of factors, and thus different and potentially incomparable topics would be produced for different datasets.
>
> We agree that other methods of textual analysis would be interesting. Due to space limitations, we had to select only some of the analysis we performed, and due to the volume of data some methods had to be avoided due to the labour required to properly analyse them. For example, we considered NER to identify sentiment towards actors in the war in Ukraine, and whether entities mentioned correlated with languages available, but it was prohibitively time-consuming to properly annotate the nations and groups each entity belonged to. We chose LIWC over other sentiment detection methods because it provides extremely fine-grained results and can serve as a linguistic profile for comparison to other datasets, and is unaffected by document length.
>
> Thank you for the suggestion to use Scattertext. We don’t believe there is sufficient space to incorporate it in this paper, but will consider it in future work.
>
> Thank you for your corrections and suggestions for clarification, we will address these.
>
> ---
>
> **A**: We did explore the categories and tags of the articles, but it appears little care is taken by the sites’ operators in using them, and as a result they are quite noisy. The author information of the sites is limited, using either generic (‘Admin’) or random (‘UiXnZyvH’) names and no useful profile data. Closer analysis of author data was conducted on a preliminary scrape - we hypothesised there may be a variation in post language or posting times for each author, but none was found. All metadata is included in the dataset, permitting future analysis by others.
>
> **B**: There appear to be no other periods of time with significant reductions in activity, as there was for the New Year and Christmas period. We suspect this is because this period is 8 continuous days, whereas other public holidays are single, isolated days.
>
> **C**: We will add a section on additional potential future work.

---

### Official Review · Reviewer_Bm7V · 2023-08-04

**Soundness:** 4

**Excitement:**

4: Strong: This paper deepens the understanding of some phenomenon or lowers the barriers to an existing research direction.

**Missing References:**

[1] Guo, Zhijiang, Michael Schlichtkrull, and Andreas Vlachos. "A survey on automated fact-checking." Transactions of the Association for Computational Linguistics 10 (2022): 178-206.

**Paper Topic And Main Contributions:**

This paper presents a novel corpus with ~3,5k unique articles collected from Reliable Recent News and War On Fakes (between March 22-23) and their translations across seven languages (14k in total). After pre-processing, the authors provide analysis with respect to the present topics and frequent n-grams; then proceed with analyzing the publication dates as well as respective patterns. Finally, they conduct an analysis on a psycho-linguistic level using LIWC2015.

**Questions For The Authors:**

[A] Which tokenizers did you use (for which languages)? Was it Spacy (as line 90 seems to indicate)?

[B] Can you provide a gist of the factuality for the collected articles (though I understand this is only a short paper)? Do you have a rough idea (from your manual annotations) what kind of articles are present with respect to factuality (e.g., mis-citing reliable sources vs making up claims)? Maybe some automated fact checking systems could provide some initial insights [1].

Suggestions:

* The number of sentences per article would also be interesting to add (e.g., in Table 8).

**Reasons To Accept:**

This paper presents an interesting corpus and analysis of articles crawled from propaganda websites. Upon publication, the presented corpus would offer a wide range of research possibilities.

**Reasons To Reject:**

Some manual annotations (for instance, on the factuality of the articles) would have been interesting; however, the presented work fits the short paper track very well.

**Reproducibility:**

4: Could mostly reproduce the results, but there may be some variation because of sample variance or minor variations in their interpretation of the protocol or method.

**Reviewer Confidence:**

2: Willing to defend my evaluation, but it is fairly likely that I missed some details, didn't understand some central points, or can't be sure about the novelty of the work.

**Typos Grammar Style And Presentation Improvements:**

* line 148: add (Appendix) to Figure 2 to improve navigation.

---

> ### Author Rebuttal · Authors · 2023-08-28
>
> Manual annotations of the articles may provide interesting further analysis, but this is out of the scope for this paper and has prohibitive costs given the large volume of articles.
>
> Thank you for your suggestions and corrections. We agree that the number of sentences is an interesting metric, we will add this if the paper is accepted.
>
> ---
>
> **A**: The mean token count statistics for all languages used the Spacy tokenizer. We will ensure this is stated explicitly.
>
> **B**: From the fact-checks published on the two sites, and our impressions from working with the content, it appears that the sites publish a mixture of outright false, misleading (e.g. mislabelling media), and some (particularly on RRN) selective true content that is advantageous to their agenda. The sites appear to follow a ‘quantity over quality’ approach, as PolitiFact identified instances of mutually conflicting stories.
>
> We agree that trialling this data on automated fact-checking systems would be interesting, but our initial impressions are that it may struggle with this data. Given the subject matter, identifying the truth is difficult and likely requires extensive domain knowledge and investigation, and as the true content appears to be written in the same style as the false content, approaches that look at credibility signals will be ineffective. Due to these difficulties, we believe a proper evaluation would be overly complex and out-of-scope for this paper.

---

### Meta-Review · Area_Chair_v6Ue · 2023-09-09

**Recommendation:** 4

**Metareview:**

This submission offers a straightforward short dataset paper, which collects 3,447 articles from two state-backed fake news websites (WarOnFakes and Reliable Recent News) using the WordPress API. The authors also perform rudimentary exploratory analysis on the dataset, by analyzing topics and article frequencies. Additionally, they analyze the dataset by comparing the distributions of LIWC words on the fake news sites to the New York Times, and counting the most frequent n-grams by month.

All reviewers offered positive assessments of the paper, although the reviews were fairly brief (perhaps because the paper was very straightforward). All three reviewers awarded an excitement score of 4, and seemed enthusiastic about the work. One noted that the submission offered an “interesting corpus” which would “offer a wide range of research possibilities.” They also pointed out that the paper “fits the short paper track very well.” Another suggested that “I think the dataset can be very valuable” and that they “found the analysis interesting, although more from a sociological, cultural perspective than from an NLP point of view.”

Because all reviewers were excited about the contribution, the only questions surrounding this submission have to do with soundness. One reviewer awarded a soundness score of 4, and the others awarded a soundness score of 3.  However, looking more closely, the reviews did not surface any major issues surrounding soundness. Therefore, there do not appear to be concerns about the soundness of this work.

- Reviewer Bm7V initially awarded a soundness score of 3, but changed it to 4 after the authors answered a question about tokenization during the rebuttal period.
- Reviewer q5Fz awarded a 3 for soundness, but did not list any concerns about soundness in their reasons to reject. Instead, they suggested that the paper might be improved if the authors include a discussion of prior scholarly study of propaganda.
- Finally, reviewer Rv3e awarded a soundness score of 3, and said they were concerned about fine-tuning during the neural topic modeling analysis. However, based on their comment, it seemed as if they actually may have been concerned about how the authors performed hyperparameter tuning. The authors replied to Rv3e with information about hyperparameter selection, which seemed to satisfy Rv3e. Rv3e thanked the authors for the additional details but did not change their score.

---

### Decision · Program_Chairs · 2023-10-07

**Decision:**

Accept-Main

**Comment:**

This submission offers a straightforward short dataset paper, which collects 3,447 articles from two state-backed fake news websites (WarOnFakes and Reliable Recent News) using the WordPress API. The authors also perform rudimentary exploratory analysis on the dataset, by analyzing topics and article frequencies. Additionally, they analyze the dataset by comparing the distributions of LIWC words on the fake news sites to the New York Times, and counting the most frequent n-grams by month.

All reviewers offered positive assessments of the paper, although the reviews were fairly brief (perhaps because the paper was very straightforward). All three reviewers awarded an excitement score of 4, and seemed enthusiastic about the work. One noted that the submission offered an “interesting corpus” which would “offer a wide range of research possibilities.” They also pointed out that the paper “fits the short paper track very well.” Another suggested that “I think the dataset can be very valuable” and that they “found the analysis interesting, although more from a sociological, cultural perspective than from an NLP point of view.”

Because all reviewers were excited about the contribution, the only questions surrounding this submission have to do with soundness. One reviewer awarded a soundness score of 4, and the others awarded a soundness score of 3.  However, looking more closely, the reviews did not surface any major issues surrounding soundness. Therefore, there do not appear to be concerns about the soundness of this work.

- Reviewer Bm7V initially awarded a soundness score of 3, but changed it to 4 after the authors answered a question about tokenization during the rebuttal period.
- Reviewer q5Fz awarded a 3 for soundness, but did not list any concerns about soundness in their reasons to reject. Instead, they suggested that the paper might be improved if the authors include a discussion of prior scholarly study of propaganda.
- Finally, reviewer Rv3e awarded a soundness score of 3, and said they were concerned about fine-tuning during the neural topic modeling analysis. However, based on their comment, it seemed as if they actually may have been concerned about how the authors performed hyperparameter tuning. The authors replied to Rv3e with information about hyperparameter selection, which seemed to satisfy Rv3e. Rv3e thanked the authors for the additional details but did not change their score.